# Tests of the Oddity Effect Hypothesis in mixed-species parid flocks

Zaharia A. Selman[1,2*], Eric K. Frazier[1,2], Colton B. Adams[1,3,4], Todd M. Freeberg[1,2,3]

1 Department of Psychology, University of Tennessee, Knoxville, Tennessee, United States of America, 2 Collaborative for Animal Behaviour, University of Tennessee, Knoxville, Tennessee, United States of America, 3 Department of Ecology and Evolutionary Biology, University of Tennessee, Knoxville, Tennessee, United States of America, 4 Department of Psychology, University of Michigan, Ann Arbor, Michigan, United States of America

* zselman@vols.utk.edu

## Abstract

Although there are major benefits of group membership, there might be severe costs to being in a group for phenotypically rare individuals. Whereas the role of rarity in antipredator behavior is well-documented in fish species, there is little empirical evidence on how rarity affects this behavior within mixed-species avian groups. Understanding this is important for clarifying how various antipredator behaviors function in different social contexts. The Oddity Effect Hypothesis predicts that predators will choose phenotypically rare individuals within groups, and as a response to their oddity, these prey individuals should behave as inconspicuously as possible, often by delaying signaling. Here we examined the role of rarity in data taken from and analyzed separately in two different, published, field experiments. We measured the latency to call in mixed-species flocks with one versus two or more individuals of Carolina chickadees (*Poecile carolinensis*), tufted titmice (*Baeolophus bicolor*), or white-breasted nuthatches (*Sitta carolinensis*) after a predator model was presented. We also tested two alternative hypotheses, the 'probability of calling' and 'recruitment' hypotheses. In support of the Oddity Effect Hypothesis, we found evidence that single individuals took longer to call: chickadees in the first experiment, using a screech owl (*Megascops asio*) model, and titmice and nuthatches in the second experiment, using a Cooper's hawk (*Accipiter cooperii*) model. For our alternative hypotheses, we found no evidence of shorter call latencies with more conspecifics in flocks due to simple probabilities of calling and no evidence of shorter calling latencies with fewer conspecifics due to increased motivation to recruit conspecifics. Our results lend support to the Oddity Effect Hypothesis, though we urge caution due to our small sample sizes.

**Data availability statement:** All relevant data are within the paper and its Supporting information files.

**Funding:** The author(s) received no specific funding for this work.

**Competing interests:** The authors have declared that no competing interests exist.

## Introduction

Animals use a wide range of anti-predator behavior patterns, including avoidance and aggressive defense behaviors. Variation in antipredator behavior has been studied in many species and environmental contexts [1–6]. Group formation is one such tactic that provides benefits of group effort in spotting and mobbing predators [7–10] or dilution through the greater number of individual members [11–13]. Groups can often deter predators by creating confusion and making it difficult for the predator to focus on any one individual, but the costs and benefits may change when an individual is rare in the group.

Theory, modeling, and experiments argue that individuals should avoid groups when they are phenotypically rare, as their rarity potentially draws attention to themselves and therefore diminishes the dilution or confusion effects [14–17]. For example, a series of controlled aquaria experiments found that predators were indeed more likely to prey upon rare species of juvenile fish in mixed-species assemblages [18]. Considering the benefits of being in a social group, however – such as communicating to each other about food, group cohesion, predators, etc. – rare individuals might still be motivated to join mixed-species groups. In these groups of animals, the Oddity Effect Hypothesis predicts that predators will select phenotypically rare individuals and so when an individual is the only member of its species at a particular time and place, it becomes a preferred target for predators [13]. Since signaling can increase detectability by predators, one major way single prey individuals in mixed-species groups can reduce the risk of predation and remain inconspicuous is to delay communication or signaling.

In acoustic communication, animals can often avoid predation by suppressing or delaying their singing or calling. For example, bladder grasshoppers (*Bullacris unicolor*) in northern and southern regions of South Africa modified their calling behavior differently depending on bat echolocation behaviors. Northern grasshoppers avoided calling during periods of bat echolocation, while southern grasshoppers did not [19]. Bats in the northern regions were mostly active early in the night, whereas southern bats were active throughout the night, an environmental variable affecting the grasshopper's ability to delay their signaling. Additionally, playbacks of predator calls reduced songbirds' singing behavior during dawn and delayed their signaling if they did sing, affecting their perceived risk of predation during that time [20]. Although there are numerous benefits to signaling in flocks, rarity may increase the costs sufficiently to cause a reduction in the signaling of the odd individual in a flock.

A commonly occurring and well-studied mixed-species avian group involves Carolina chickadees (*Poecile carolinensis*), tufted titmice (*Baeolophus bicolor*), and white-breasted nuthatches (*Sitta carolinensis*). These North American songbirds exhibit significant range and niche overlap from the midwestern to the southeastern United States [11,21–23]. Mixed-species flocks (MSFs) of these individuals usually form in the winter when predation risk is high [24]. These flocks differ in size and composition of conspecifics and heterospecifics, making them a suitable model for studying the effect of group composition on specific antipredator behavior [25–28].

Information about predators in MSFs usually flows from nuclear, leader species (chickadees and titmice) to information scrounger, satellite species (nuthatches) that join these flocks [29,30]. Chickadees and titmice use complex and multipurpose *chick-a-dee* calls in a wide range of contexts [31]. These calls are produced by both sexes throughout the year and comprise sequences of distinct note types that result in many unique call types, potentially conveying enormous amounts of motivational and behavioral information about each individual [31,32]. This call serves as an important signal in flock cohesion, identity, and, more importantly to this study, alarm and predator mobbing contexts [8,28,33–35].

When nuthatches join MSFs of chickadees and titmice, they can benefit from the predator vigilance and increased foraging ability gained by eavesdropping on *chick-a-dee* calls [25,36–38]. Although nuthatches are satellite species that can exploit these benefits, they also produce calls that contain relevant information for the flock to use [30]. Nuthatches do not have a *chick-a-dee* call system, but they commonly use 'quank' and 'hit-tuck' calls, two call types that are used by both sexes throughout the year in a wide range of social contexts [21,39]. Variation in these calls may be related to signaling about predators or other environmental threats [11,40,41].

To investigate how flock composition mediates calling response in visual predator contexts, we sought to determine how long it took a single Carolina chickadee, tufted titmouse, or white-breasted nuthatch in MSFs to call in comparison to flocks with two or more conspecifics in MSFs, respectively, in two different field studies. Our data sets for these studies come from two previously published projects that assessed potential roles of habitat density, traffic noise, and MSF size and composition on anti-predator calling and seed-taking behavior [28,42]. In the present study, for each species we assessed the latency to call after being presented with a predator model. Aligned with the Oddity Effect Hypothesis, we predicted that when an individual was the only one of its species present in the flock, as a response to oddity, it would have a longer latency to call in the presence of the model compared to when there were two or more conspecifics within the MSF.

Alternatively, any single individual may call with relatively low probability under conditions of risk, so call latencies for each species may increase with decreases in the number of individuals of those species. Under this 'probabilistic calling' hypothesis, we would expect a clear negative relationship between the number of individuals and the latency to call for each species. An alternative hypothesis to probabilistic calling relates to flock member recruitment. In the 'recruitment' hypothesis, individuals may be *more* motivated to call when they are rare if increasing their calling serves to recruit conspecifics into a flock, as seen in assembly near a newly found food resource [26] or mobbing behavior [8]. Chickadees are known to produce *chick-a-dee* mobbing calls once a predator has been seen and these calls can recruit others to harass and deter the predator from the area [8]. Additionally, birds may also call to recruit others to the area to avoid the oddity effect as the recruitment may reduce the likelihood that one species will be located by and preyed upon by the predator. Thus, the recruitment hypothesis makes the opposite prediction of both the probabilistic calling and the response to oddity predictions.

## Methods

### General

We tested for responses to oddity and these two alternative hypotheses in two experiments using two different predator models in eastern Tennessee at the University of Tennessee Forest Resources, AgResearch, and Education Center (36.11° N, 84.20° W) [28,42]. We conducted each experiment once at 36 feeder sites, separated by a minimum distance of 375 meters (m) to help ensure that each feeder site was independently sampled and in a different flock's territory [21].

Each feeder site consisted of a wooden feeding platform (25 cm × 40 cm × 2 cm) attached to a steel pole driven into the ground, so the platform was roughly 1.5 m above ground level. Beginning in the fall before the year of the study and continuing through the end of each study, each feeding platform was stocked every 10–14 days with 50 g of mixed bird seed to encourage flock use. Our two experiments involved the placement of an avian predator model on or near these feeding

platforms (Fig 1). In both experiments, a flock had to contain at least one Carolina chickadee and at least one tufted tit-mouse observed taking seed from the feeder for us to begin recording a trial at that site. Data from these two experiments were used to assess the Oddity Effect Hypothesis in the context of predator models of different threat levels [7–8].

### Screech owl model experiment

We used a realistic plastic model of an eastern screech owl (*Megascops asio*) that was used in the previously published study [28], where more detailed methods can be found. Although screech owls primarily hunt at night, they can hunt in dawn and dusk hours, and the birds in our MSFs typically react strongly to screech owl stimuli [11,23]. This experiment

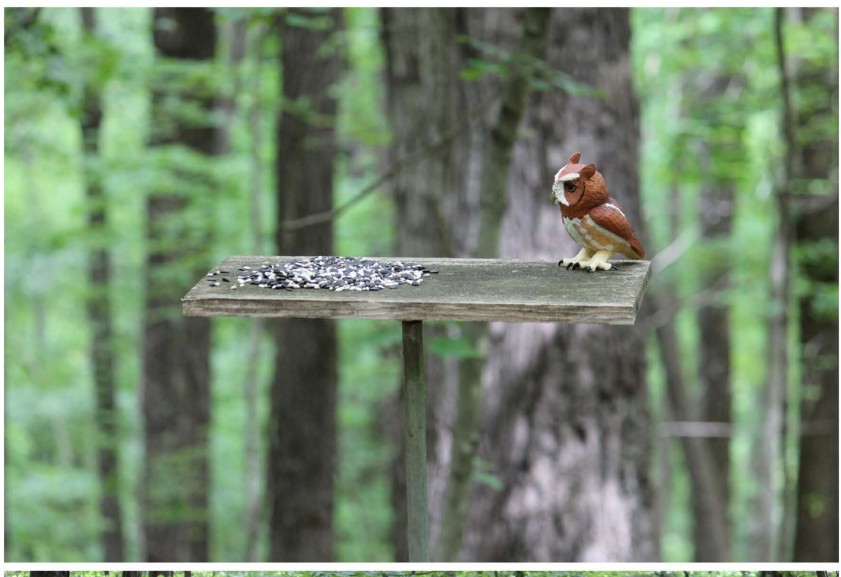

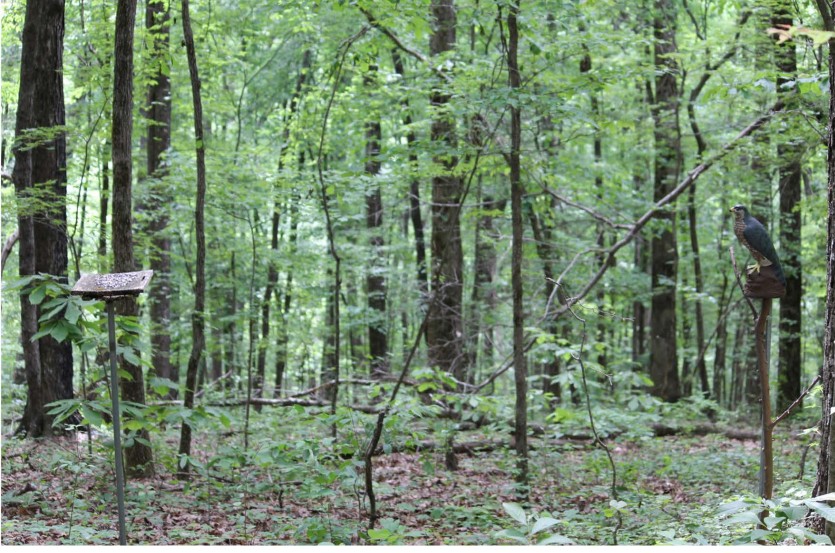

**Fig 1. Experimental Set-Up.** Example of experimental set-up at the feeder platforms for the screech owl (top; [28]) and Cooper's hawk (bottom; [42]) experiments. The microphone (out-of-view) that recorded bird calls was on the ground and faced the feeder. For the Cooper's hawk experiment, the predator model was mounted on an artificial tree.

ran from 05 February to 11 March 2022 (S1 File). Each trial was 20 min long. Observers moved in the same way during the pre-stimulus and stimulus periods to avoid affecting the birds' behaviors as much as possible. After setting up the equipment, the observer touched the feeder before retreating to the observation chair hidden behind vegetation 10 m away to simulate the same movements as placing the predator model on the platform in the stimulus period [28,43]. The observer conducted the pre-stimulus period for roughly 10 min to determine the number of individuals of each species. We assessed the maximum number of individuals of each species detected within 20 m of the feeder at any one time, which has been shown to be highly positively correlated with the actual number of individuals present at the site [21]. Following this period, the observer walked to the feeder again and placed the owl model on a corner of the feeder platform so that its head and body were oriented toward the seed present on the platform (Fig 1). The observer then returned quickly to the previous observation space. Calls of the three species were recorded using a Sennheiser ME-64 microphone set up near the feeders connected to a Marantz PMD-660 digital recorder and saved as uncompressed.wav files.

## Cooper's hawk model experiment

Our data set for the second study came from a previously published experiment with a Cooper's hawk (*Accipiter cooperii*) model that was conducted from 11 January to 29 February 2024, with data collection occurring between 0800 and 1500 EST (S1 File) [42]. Feeding platforms were stocked with 50 g of seed on the day of trials. Once a flock was regularly taking seed from a feeding platform, we set up the recording equipment as described above. The observers sat hidden behind vegetation at least 15 m away from the feeding platform (after [21]) and used another microphone to record observer commentary into the second recording channel.

Our predator stimulus was a realistic plastic model of an eastern Cooper's hawk [21]. Chickadees, titmice, and nuthatches respond strongly to real and model Cooper's hawks. After the recording equipment set-up was complete, the observer mounted an artificial tree (predator model mount) into the ground 3 m from the feeding platform at roughly the same height as the feeding platform and directly in line with the microphone's direction (Fig 1). The artificial tree was a natural wooden branch attached to a 1.5 m length of rebar driven into the ground to make the artificial tree stand up. After setting up the equipment, touching the predator model stand, and returning to the observation space, there was a 15 min pre-stimulus period. During this time, the birds were habituated to the artificial tree, and we aimed to determine the number of individuals of each species present as described above.

Following the pre-stimulus period, the observer walked back to the model stand and placed the Cooper's hawk model on top of the predator model mount. The model was always placed on the mount so that the hawk's eyes were directly facing the feeder, as facial orientation is known to produce stronger antipredator behavior in these species [27,44]. The observer then returned immediately to the observation space and audio-recorded the mixed-species flock activity for 15 min. For both experiments, birds in our observation areas usually avoid the feeding platform when observers get too close (during the onset of both the pre-stimulus and stimulus periods) but generally return to the feeding platform quickly once observers return to their observation distance.

## Data coding and analysis

Our key data were the first calls produced by chickadees and titmice – and nuthatches, if they were present – in each flock. For chickadees and titmice, we coded/identified only *chick-a-dee* calls, as these are the most common calls produced by these species near our feeders and have been the calls of interest in studies of antipredator and mobbing behavior in these species [31]. Alarm calls and gargles for chickadees and alarm calls and agonistic 'squeal' calls for titmice were relatively rare in our data recordings. For nuthatches, we coded their 'quank' and their 'hit-tuck' calls. We obtained data at all sites for the screech owl model experiment. Due to researcher error or equipment malfunction in the Cooper's hawk model experiment, calls could not be coded from six sites, so our sample size was limited to 30 flocks for calling behavior for chickadees and titmice and 27 flocks for nuthatches.

Our dependent variable in this study was the latency to call in the context of the avian predator model. Kolmogorov-Smirnov tests indicated significant departures from the normal distribution, and the standard deviations were all larger than the means, for all data sets. Because of the extreme non-normality of the data, we used non-parametric tests in this study (though we note that GLM analyses using the negative binomial distribution obtained qualitatively similar results). To test for responses to oddity, cases of one individual being present in the flock were compared to cases of two or more individuals being present in the flock, and potential differences were assessed using two-tailed exact Mann-Whitney U tests. The prediction under this hypothesis is that a single individual of a species in a flock will take significantly longer to call in comparison to cases of two or more conspecifics in a flock. To test the probability of calling alternative hypothesis, we assessed correlations between the number of conspecifics in the flock and the latency for the first individual to call. We excluded instances from the "Oddity Effect" data when just one individual of a species was present. The prediction under this hypothesis is a significant *negative* correlation between calling latency and the number of conspecifics. In contrast, to test the recruitment alternative hypothesis, we assessed correlations between the number of conspecifics in the flock and the latency for the first individual to call. Therefore, the prediction under this hypothesis is a significant *positive* correlation between calling latency and the number of conspecifics. We used non-parametric Spearman correlations to test these hypotheses, and all analyses were performed using IBM SPSS Statistics Version 27.

## Ethics statement

This field experiment was conducted under approved Institutional Animal Care and Use Committee protocol #1248 at the University of Tennessee, Knoxville. Our experiments were observational and did not involve the handling of any individuals. Although predator stimuli can be temporarily stress-inducing, predator encounters are a normal occurrence, and thus these experiments are considered benign.

## Results

In the screech owl experiment, flocks consisted of a mean ± SD (median, range) of 3.83 ± 1.42 (4, 1–6) chickadees, 3.03 ± 1.5 (3, 1–6) titmice, and 1.67 ± 0.83 (2, 0–3) nuthatches. The average total flock size (the number of individuals of these three species plus any other satellite species present at or near the feeders) was 9.61 ± 2.75 (10, 4–16) birds. In the Cooper's hawk experiment, there were 3.03 ± 1.61 (3, 1–8) chickadees, 2.69 ± 1.26 (2, 1–6) titmice, and 1.39 ± 0.96 (2, 0–3) nuthatches. The average total flock size was 9.17 ± 2.89 (9, 5–16) birds.

In the screech owl experiment, two of the 36 flocks containing chickadees had just one chickadee, six of the 36 flocks containing titmice had just one titmouse, and only one of the 25 flocks containing nuthatches had just one nuthatch (for this reason, we did not run statistical analyses on nuthatches). Compared to flocks with two or more chickadees, sole chickadees in mixed-species flocks called with a longer latency (Mann-Whitney U = 1, n1 = 2, n2 = 34, two-tailed p = 0.006; Fig 2a). We did not detect a difference in calling latency for tufted titmice (Mann-Whitney U = 79, n1 = 6, n2 = 30, two-tailed p = 0.66; Fig 2b). Testing the probability of calling hypothesis, we detected no significant negative relationship between calling latencies and number of individuals for chickadees (Spearman's ρ = −0.121, N = 34, p = 0.495) or titmice (ρ = −0.145, N = 30, p = 0.445). Testing the recruitment hypothesis, we detected no significant positive relationship between calling latencies and number of individuals for chickadees (Spearman's ρ = −0.258, N = 36, p = 0.159) or titmice (ρ = −0.162, N = 36, p = 0.344).

In the Cooper's hawk experiment, two of the 30 flocks containing chickadees had just one chickadee, two of the 30 flocks containing titmice had just one titmouse, and five of the 27 flocks containing nuthatches had just one nuthatch. We did not detect a difference in calling latency for Carolina chickadees (U = 10, n1 = 2, n2 = 28, two-tailed p = 0.159; Fig 3a). Compared to flocks with two or more titmice, sole titmice in mixed-species flocks called with a longer latency (Mann-Whitney U = 2, n1 = 2, n2 = 28, two-tailed p = 0.016; Fig 3b). Compared to flocks with two or more nuthatches, sole nuthatches in mixed-species flocks also called with a longer latency (U = 12, n1 = 5, n2 = 22, p = 0.005; Fig 3c). Testing the

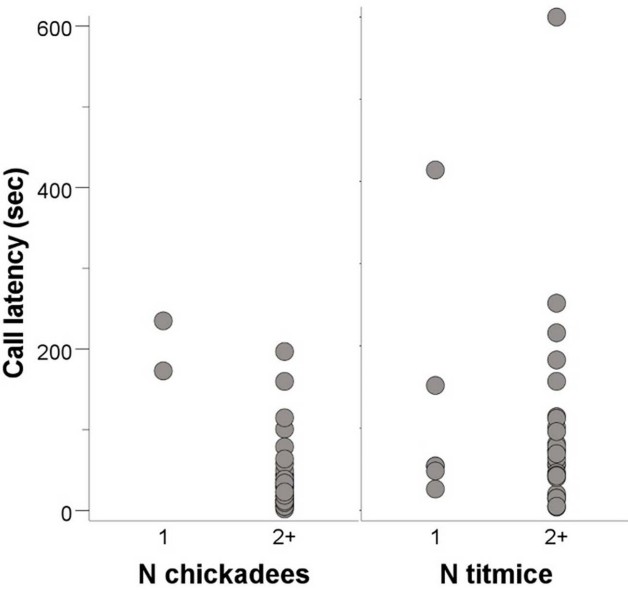

**Fig 2. Call latency in screech owl experiment.** Calling latencies for Carolina chickadees (left) and tufted titmice (right) in the screech owl experiment. Only data for chickadees were statistically significant.

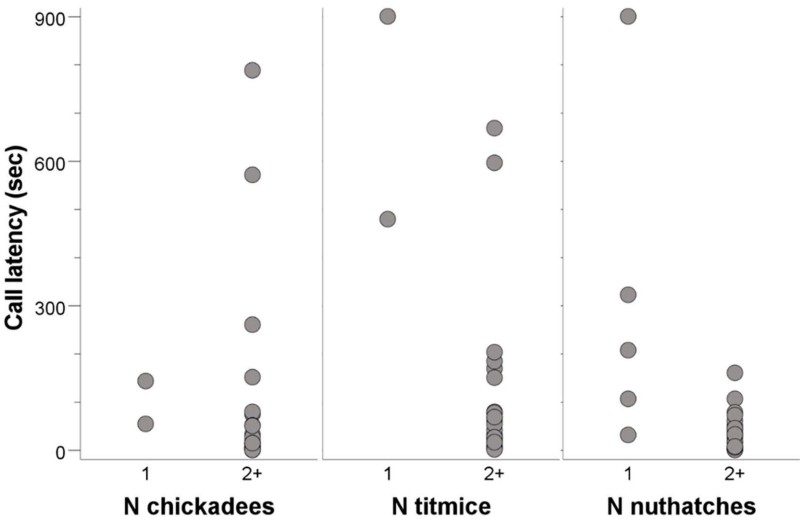

**Fig 3. Call latency in Cooper's hawk experiment.** Calling latencies for Carolina chickadees (left), tufted titmice (middle), and white-breasted nuthatches (right) in the Cooper's hawk experiment. Only data for titmice and nuthatches were statistically significant.

probability of calling hypothesis, we detected no significant negative relationship between calling latencies and number of individuals for chickadees (Spearman's $\rho=-0.190$, N = 28, p = 0.332), titmice ($\rho=+0.331$, N = 28, p = 0.085), or nuthatches ($\rho=-0.042$, N = 22, p = 0.854; 19 instances of N = 2 and 3 instances of N = 3 nuthatches). Testing the recruitment hypothesis, we detected no significant positive relationship between calling latencies and number of individuals for chickadees (Spearman's $\rho=-0.301$, N = 30, p = 0.106), titmice ($\rho=+0.079$, N = 30, p = 0.679), or nuthatches ($\rho=-0.464$, N = 27,

p = 0.015). Although we detected a significant relationship between calling latency and number of nuthatches, the prediction for recruitment was not supported since the negative direction of the effect in nuthatches is opposite of the prediction for recruitment.

## Discussion

Although the Oddity Effect Hypothesis is widely cited, the evidence for it is comparatively limited [45,46]. The argument and findings for the Oddity Effect are that predators focus their attacks on rare individuals in a group, perhaps because those individuals are more perceptually salient [47,48]. For example, experimentally phenotypically altered prey species tended to be preyed upon more by fish predators if they were relatively rare [16,49]. When exposed to predation cues, mixed-species shoals of fish were more likely to break up into single-species groups [50]. A key prediction of the Oddity Effect Hypothesis is that single individuals in mixed-species flocks (MSFs) should delay signaling in response to predator stimuli – as a response to oddity – in comparison to individuals having one or more conspecifics in those flocks. In this study, we found that single individuals in MSFs took longer to call in response to the presence of a predator: chickadees (but not titmice) in the screech owl experiment and titmice and nuthatches (but not chickadees) in the Cooper's hawk model experiment.

To our knowledge, this study provides some of the first empirical evidence in support of the Oddity Effect Hypothesis in mixed-species flocks of birds. Most research on the Oddity Effect Hypothesis on mixed-species groups has focused on mixed-species shoals of fish. Additionally, previous avian studies either found little support for the hypothesis [51] or focused on single-species flocks [52]. Researchers have investigated how the number of conspecifics and heterospecifics influenced calling behavior in Carolina chickadees and tufted titmice when presented with a simulated risk [51]. They did not find support for the Oddity Effect from either species, but they also did not have direct comparisons of flocks with a sole individual versus two or more individuals of a particular species, as we did in the current study. Other researchers have observed the Oddity Effect in Goshawks (*Accipiter gentilis*), which overcome the confusion effect by preying selectively on phenotypically odd pigeons (*Columba livia*), a foraging technique linked to increased reproductive success [52]. Although our sample sizes were small, we found that birds took longer to call in mobbing-like contexts if they were the only individual of their species present in the flock.

### Screech owl experiment

In the screech owl experiment, data from chickadees (but not titmice) supported expectations for prey individuals responding to their oddity (we did not analyze data for nuthatches as described above). Latency to call is affected by a diversity of social and physical variables in the changing environments of bird species. Flock mate familiarity may be one such factor that contributed to the asymmetry in latency to call observed between these two species. Conspecific flock size has been found to greatly influence the calling latency of Carolina chickadees [28,53]. Carolina chickadees in conspecific familiar flocks housed in aviaries called quicker when presented with various predator stimuli in comparison to flocks in which individuals were not familiar with one another [54]. Additionally, conspecific calls elicited a quicker response from black-capped chickadees (*Poecile atricapillus*) than heterospecific calls [55]. This evidence supports the lack of calling response from sole chickadees in our MSFs.

Our findings also suggest differences in role characteristics within MSFs in certain contexts as titmice did not delay their signaling when they were the only individual of their species present in the MSF [11,23]. Titmice may act bolder even in the absence of conspecifics within the MSF [11,28,56]. This suggests that conspecific flock size plays a greater role in influencing calling and anti-predator behavior for certain species (e.g., chickadees) over others.

## Cooper's hawk experiment

In the Cooper's hawk experiment, data from both titmice and nuthatches supported our expectations related to oddity, though data from chickadees did not. Why we obtained different results in this experiment compared to the first is not clear, though it is clear that contextual variation and the presence of certain species can change the role of the nuclear species within MSFs. Flock composition's effect on leader species was studied in MSFs of birds in the Yungas foothill forests of the southern Andes [57]. In their study, 35 species socially behaved like leaders, but only under certain conditions. For example, the tropical parula (*Setophaga pitiayumi*) only behaved as a leader species in the absence of the brown-capped whitestart (*Myioborus brunniceps*), common bush tanager (*Chlorospingus flavopectus*), suiriri flycatcher (*Suiriri suiriri*), or white-throated tyrannulet (*Mecocerculus leucophrys*) [57]. In our study, these role changes may have varied between the two nuclear species because of the presence and perceived threat of the predator models.

Both Carolina and black-capped chickadees have been observed to alter their calling behavior based on their perceived threat of predators [7,8,34,58]. Carolina chickadees responded to calls produced in response to smaller, higher-threat predator models (e.g., eastern screech owl) than to larger, lower-threat models [34] (red-tailed hawk, *Buteo jamaicensis*). The larger size of the Cooper's hawk model compared to the screech owl model in this study could have decreased the Carolina chickadee's perceived risk, increasing their boldness and tendency to call. Conversely, the larger size of the hawk model could have served as an increased risk for the titmice and nuthatches, causing a single titmouse or nuthatch to suppress their calling behavior when they were the only member of their species in these MSFs. Additionally, black-capped chickadees responded significantly more to treatments of greater risk when predator models were 1 m (versus 6 m) away from birds [7]. Although in our study the predator model was placed on the feeding platform in the screech owl experiment, the model was placed 3 m away from the feeding platform in the Cooper's hawk experiment. This distance is within the range that has been found to invoke a mobbing response from chickadees [7,21] but likely indicates a lower threat level to chickadees than the predator model perched directly on the feeding platform. In our MSFs, more work is needed to test whether certain flock compositions and perceived predator contexts cause species to change roles and display bolder antipredator behaviors within the flock [59].

## Conclusion

Given the likely costs of joining a mixed-species group by a single individual of a species, particularly if that individual is phenotypically different, why would such an individual join the group? Studies have found that individuals of prey species prefer to join groups with individuals of similar phenotypes compared to groups with individuals that differ from their phenotype [60–62]. Despite the oddity costs, the benefits of joining a mixed-species group when one is the sole member of its species can still outweigh the predation costs of not joining and remaining solitary [16]. These benefits can accrue for many reasons, including there simply being more individuals present to detect a potential predator [3]. This is perhaps particularly true when the single individual has a specific defensive anatomy that can protect it [63] or is quite cryptic relative to the other species it joins in the mixed-species group [12]. Additionally, models predict that the Oddity Effect is stronger the larger the group, so a single individual of its species should be more likely to join a mixed-species group that is relatively small [17]. To avoid predators, single individuals of their species must balance the benefits of joining MSFs and the risks of oddity. If they do join an MSF, they may need to delay signaling in contexts of risk [12].

A limitation of our study, as mentioned above, is the relatively small sample sizes of single individuals of each species in these two field experiments. Conversely, we should note that the relative lack of single individuals of a species in these mixed-species flocks might be additional evidence in support of the Oddity Effect Hypothesis. More studies of mixed-species flocks in a greater diversity of avian species are needed. Additional studies might also help shed light on potential social mechanisms to explain the Oddity Effect in signaling behavior.

Assuming signaling behavior is primarily intended for conspecifics, there might be little motivation for an individual to call in mixed-species flocks in the absence of conspecifics, and we believe our results show this. Alternatively, in risky contexts, each individual might have a relatively low probability of signaling – if so, simply based on stochastic processes underlying calling, we might predict signaling latencies to increase with decreasing numbers of individuals of a particular species. However, our correlational analyses of latencies to call based on the number of individuals, when at least two individuals occurred in the flock, found no support for this idea. The lowest p-value we detected was for titmice in the Cooper's hawk study, but the correlation in that case was positive rather than negative – titmice tended to delay calling the *more* other titmice were in the flock. Whether signaling occurs or not between group members is known to depend on many internal, external, and/or behavioral factors (e.g., social relationships or motivation) beyond just conspecific identification and may explain why our results did not support this hypothesis [64–66].

Finally, we found no evidence for the alternative hypothesis that calling relates to recruitment of conspecifics, such that calling latencies should be positively associated with the number of conspecifics. Our only significant finding here was for a *negative* relationship between calling latencies and the number of nuthatches in the Cooper's hawk study. However, support for the recruitment hypothesis has been found for titmice and chickadees in other studies. Distress calls of tufted titmice may serve to attract conspecifics [67,68]. Additionally, in a feeding context, Carolina chickadees produced calls with greater numbers of D notes that seemingly served to recruit subsequent chickadees that came to the feeder [26]. Our findings suggest that calling behavior in all three of our focal species supports the Oddity Effect Hypothesis in the presence of a predator, but only in certain contexts. However, we repeat our note of caution. Our supportive evidence is based on a small number of individuals who were the "odd" members of their flocks. More work with larger sample sizes is needed and, ideally, experimental work that manipulates the number of individuals in mixed-species groups as well as the different contexts related to predation. Antipredator behavioral responses in studies with only visual predator stimuli have been found to differ from those with only acoustic (alarm-call playbacks and predator calls) stimuli [11,28,30,62]. In general, visual predator models seem to be more salient than models with acoustic cues, likely because of their higher certainty in localizing the predator [69]. The modality of predation risk assessment, whether visual, acoustic, or both, can drive differences in antipredator behavior. Future studies should investigate how various predator model modalities and MSF compositions affect role reversals and antipredator behaviors.

## Supporting information

**S1 File. Data Spreadsheet.**
(XLSX)

## Acknowledgments

We would like to thank members of the Special Topics in Animal Behavior Seminar, Heather Brooks, two anonymous reviewers, and members of the Comparative Communication Lab for their helpful comments and suggestions that greatly improved this manuscript.

## Author contributions

**Conceptualization:** Todd M. Freeberg.

**Data curation:** Zaharia A. Selman, Eric K. Frazier, Colton B. Adams, Todd M. Freeberg.

**Formal analysis:** Zaharia A. Selman, Eric K. Frazier, Colton B. Adams, Todd M. Freeberg.

**Investigation:** Zaharia A. Selman, Eric K. Frazier, Colton B. Adams.

**Methodology:** Zaharia A. Selman, Eric K. Frazier, Colton B. Adams, Todd M. Freeberg.

**Project administration:** Zaharia A. Selman, Eric K. Frazier, Colton B. Adams.

**Resources:** Todd M. Freeberg.

**Software:** Zaharia A. Selman, Eric K. Frazier, Colton B. Adams, Todd M. Freeberg.

**Supervision:** Todd M. Freeberg.

**Validation:** Zaharia A. Selman, Eric K. Frazier, Colton B. Adams, Todd M. Freeberg.

**Visualization:** Todd M. Freeberg.

**Writing – original draft:** Zaharia A. Selman.

**Writing – review & editing:** Zaharia A. Selman, Eric K. Frazier, Colton B. Adams, Todd M. Freeberg.

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
