## [Decision Letter · Decision Letter 0]

17 Jul 2025

Dear Dr. Selman,

Thank you for submitting your manuscript “Tests of the Oddity Effect Hypothesis in mixed-species parid flocks” to PLOS One. I regret that I am unable to accept your paper for publication in Plos One in its present form. However, I would be prepared to consider for publication a substantially revised version that takes into account the suggestions of the reviewers. Such a revised paper would be reviewed and there would be no guarantee of acceptance.

As you can see, both reviewers agree that the study could, in principle, be a meaningful contribution to our understanding of the behaviour of animals in mixed-species groups, but also raise important issue regarding the soundness of the analyses, the clarity of the rationale for the study in general and specific approaches in particular, and identify a weakness in terms of power that needs to be acknowledged and incorporated in a substantial revision of the discussion and interpretation of the findings.

In particular, R1 highlighted how the small sample size is not sufficient to draw robust conclusions, and since this aspect cannot be fixed post-hoc, the interpretation of the findings need to be consequently toned town to acknowledge this aspect.

Additionally, R2 raised important concerns on the statistical approach used, and recommends the used of mixed models as a more suitable way to take into consideration multiple potential factors that could affect your analysis. I urge you therefore to revise the analyses and the discussion in light of the new findings.

Finally, both reviewers have a number of other comments (detailed below), which you should address in your revision.

With best regards,

Valeria Mazza

Acedemic Editor

PLOS ONE

We look forward to receiving your revised manuscript.

2. Please include captions for your Supporting Information files at the end of your manuscript, and update any in-text citations to match accordingly. Please see our Supporting Information guidelines for more information: http://journals.plos.org/plosone/s/supporting-information .

Reviewers' comments:

Reviewer's Responses to Questions

**Comments to the Author**

1. Is the manuscript technically sound, and do the data support the conclusions?

Reviewer #1: Partly

Reviewer #2: Partly

2. Has the statistical analysis been performed appropriately and rigorously?

Reviewer #1: No

Reviewer #2: No

3. Have the authors made all data underlying the findings in their manuscript fully available?

Reviewer #1: Yes

Reviewer #2: Yes

4. Is the manuscript presented in an intelligible fashion and written in standard English?

Reviewer #1: Yes

Reviewer #2: No

Reviewer #1: Manuscript ID: PONE-D-25-21798

Title: Tests of the Oddity Effect Hypothesis in mixed-species Parid flocks

General comments

This is a manuscript regarding a very interesting topic, not very investigated. However, I found some weaknesses about clarity of methods and results. In particular, you tested the oddity effect hypothesis with very small sizes (i.e. N = 2) for sole birds and although you used non parametric tests, you should be very cautious to draw conclusions.

Specific comments

Methods

1. Line 150-157 and 178-182: if I understand correctly, in both experiments, you had to wait 10-15 minutes before showing the predator model mount, in which you count birds in the flock. Then you walked to the feeder, placed the model, and returned to the observation place. Don’t you think that the operator could have affected in some way the birds behavior? Do birds remained there, at the feeder, or flew away when you walked to place the predator models? I think that it is important to write something about it, either if operator was disturbing or not.

2. Line 157: how far was the observation place in the screech owl experiment? Was the observer hidden or in plain sight? Please, add these details.

Results

3. Lines 205-208: so, to test both the “probability of calling” and the “recruitment” hypotheses, you correlated the same things, i.e. number of conspecifics in the flock vs the latency for the first individual to call. What differed was the prediction, right? Aren’t these hypotheses in contrast? When a bird is rare in a mixed flock either could be silent to doesn’t draw attention (probability of calling hypothesis) or be loud to recruit conspecifics (recruitment hypothesis). I suggest to say something about this, also in discussion, if there is some threshold that lead to one or the other hypothesis, and why a rare bird choose to be quite or loud.

4. Lines 223-225 and 235-242: as I understand, you compared latencies of 2 vs 34 chickadees flocks in the screech owl experiment, latencies of 2 vs 28 chickadees flocks in the hawk experiment, and latencies of 2 vs 28 titmice flocks in the hawk experiment. Therefore, even if you used non-parametric tests, I suggest to be very cautious to draw some conclusions with these results for the oddity effect hypothesis.

Discussion and Conclusions

5. As I said in the previous comment, because of the small sample size you should be very careful to draw conclusions about the oddity effect hypothesis, as you said as well in conclusions. However, you seemed very convinced of these results (see lines 263-265, 277-279, 282, 299), but I suggest to rephrase using a more cautious tone.

Reviewer #2: General comments: I think this is a nice study, and a good addition to the growing body of literature that is trying to dig into individual behavioral changes in flocking contexts. I do, however, think that the abstract, introduction, and methods could all use some restructuring to make both the question, statistics and field methods clearer. In particular, it is hard to understand exactly what data were taken, whether the data between the two experiments is different, and what the purpose of the two experiments was. I also would strongly consider the authors changing their statistical analysis to take advantage of GLMs and the fact that they can incorporate multiple factors and interactions. I think this reframing in a way that unifies the data and the variability present in the mixed-species flocks may significantly impact the discussion as well.

Abstract:

Line 15 - This first sentence is confusing and, as it stands, could be interpreted multiple ways. I would advise reframing this and maybe swapping the major points of the first two sentences in the abstract. First, describe rarity, then describe its role in predation.

Line 25 - The description of the experiment here is also a bit confusing. Given this is the abstract, there isn't much space to explain further, but the details added here don't clarify your findings and instead pose several questions. Why two experiments? How did they differ? Were they analyzed together or separately? Given that (after going through the rest of the paper) the data for the two experiments are different yet are addressing the same question in very similar ways I would just simplify this and state that you had evidence that at least two species (chickadees and titmice) were less likely to call (alarm call?) in response to predators when alone as compared to with conspecifics. I would then maybe use that tiny bit of extra space to explain further the comment below.

Line 27-28 - I also had trouble following this explanation of ruling out alternative hypotheses. Are these your alternatives, or is this what you found? Maybe stating that you tested both the oddity effect and these two alternative hypotheses in one sentence earlier in the abstract would help clarify.

Introduction:

Line 65 - Somewhere here I think it is important to state that mixed-species groups are often signaling to each other - transmitting information about food or predators etc. This will set up the reasoning below that it isn't always good for specific (rare) individuals to call.

Line 67 - might be better here to emphasize that signaling can increase predation by making individuals more conspicuous, prior to describing how a reduction in signaling can mitigate this.

Line 75 - 76 - sentence here about attracting mates and deterring competitors feels out of place. somewhere here, it might be better to lean into the mixed-species group stuff and talk about how there may be benefits to signaling in flocks, but that rarity may interact with appearance /vocal distinctiveness to make it not always beneficial and thus a contributing factor to flock formation, function, and individual behavior

Line 86 - mixed species should have a dash in it

Line 105 - single conspecifics within heterospecific flocks? And multiple conspecifics within heterospecific flocks? Assuming yes, but this should be clear.

Line 117 - 121 - describing the hypothetical probabilities here seems unnecessary. I would in stead use this space to describe your second alternative hypothesis, perhaps describing the recruitment of birds to mobbing flocks more explicitly. May also expect that birds would call to recruit in order to avoid the oddity effect.

Methods:

General comment - I'd recommend putting a sentence in near the front of the methods here clearly stating you conducted two experiments using two different predator stimuli. The first you mention that there were two is "Our two experiments involved..."

Also would be helpful to clarify why you used these two different experiments. Is it because these two predators represent different threat levels?

Line 146 - It is implied, but not clear, that you also used this same plastic screech owl that was used in the other study. Also might be good to state that in the previous work birds responded to it like a predator, as many birds get habituated to plastic stimuli.

Line 152 - are you assuming all the individuals remained in the area following the placement of the stimuli? Did you account for any individual departures driven by the presence of the observer with the stimulus?

Line 157 - Was the call type recorded? Also, assuming you recorded what species was producing each call.

Line 184 - What do you mean by "coded"? Identified? Did you record the time to the first call as well as the number of each call type?

General method question: Were each of the experiments only conducted once at each of the 36 feeder sites?

Line 195 - explain how the data were non-normal

Line 194 - This paragraph could use a bit more structure. For example, consider telling us how many analyses you ran to answer what questions. Even having a list of 1, 2, 3 might help. It is also not clear what was analyzed for which data set. Were they all compiled?

General methods question - given that there is likely a variable number of both heterospecifics and conspecifics in each of these experimental flocks, I'd recommend using GLMs or a similar analysis that would allow you to incorporate both the number of conspecifics and heterospecifics and the potential interaction between them into your analysis. You could also include species and put all the species into the same analysis.

Line 212 - The word ethically here isn't informative. Maybe instead say "Experiments were observational and did not involve the handling of any individuals. While predator stimuli can be temporarily stress-inducing, predator encounters are a normal occurrence, and thus these experiments are considered benign."

Line 216 - consider rephrasing this: "In the screech owl experiment, flocks consisted of..." Can you also include the mean flock size over all for each experiment?

Line 249 - you say there was no significance, but report a p value of 0. 015 for nuthatches. is this a typo?

**Do you want your identity to be public for this peer review?** For information about this choice, including consent withdrawal, please see our Privacy Policy

Reviewer #1: No

Reviewer #2: No

---

## [Author Response · Author response to Decision Letter 1]

3 Nov 2025

Response to Reviewers – Plos ONE (PONE-D-25-21798)

Tests of the Oddity Effect Hypothesis in mixed-species parid flocks

Dear Dr. Mazza:

Thank you for giving us the opportunity to revise and resubmit our manuscript to

address the helpful criticisms and questions of the reviewers. We have copied and

pasted the entire set of comments of the reviewers, which we include below. For each

comment, we briefly describe in red text how it was addressed and, if relevant, where

the changes can be found (by line number) in the clean, non-track-changes manuscript.

+++++++++++++++++++++++++

Thank you for submitting your manuscript “Tests of the Oddity Effect Hypothesis in mixed-species parid flocks” to PLOS One. I regret that I am unable to accept your paper for publication in Plos One in its present form. However, I would be prepared to consider for publication a substantially revised version that takes into account the suggestions of the reviewers. Such a revised paper would be reviewed and there would be no guarantee of acceptance.

We thank you and the reviewers for the opportunity to resubmit our manuscript, regardless of the end decision!

Thank you for these comments, they have been addressed in the updated manuscript!

***Reviewer 1:***

Is the manuscript technically sound, and do the data support the conclusions?

Reviewer #1: Partly

- We thank the reviewer for the comment and hope we have expanded our manuscript to fully meet the requirements for this specification.

Has the statistical analysis been performed appropriately and rigorously?

Reviewer #1: No

- Thank you for this comment, it has been addressed in the updated manuscript in Lines 194-212. We opted to keep the non-parametric approach given its simplicity and conservative nature for studies with small Ns. We note below and in the text, however, that GLM analyses resulted in similar outcomes.

Have the authors made all data underlying the findings in their manuscript fully available?

Reviewer #1: Yes

- We thank the reviewer for the supportive comments!

Is the manuscript presented in an intelligible fashion and written in standard English?

Reviewer #1: Yes

- We thank the reviewer for the supportive comments!

General comments

This is a manuscript regarding a very interesting topic, not very investigated. However, I found some weaknesses about clarity of methods and results. In particular, you tested the oddity effect hypothesis with very small sizes (i.e. N = 2) for sole birds and although you used non parametric tests, you should be very cautious to draw conclusions.

- We thank the reviewer for the supportive comments and hope we have addressed all concerns adequately below!

Specific comments

Methods

1. Line 150-157 and 178-182: if I understand correctly, in both experiments, you had to wait 10-15 minutes before showing the predator model mount, in which you count birds in the flock. Then you walked to the feeder, placed the model, and returned to the observation place. Don’t you think that the operator could have affected in some way the birds behavior? Do birds remained there, at the feeder, or flew away when you walked to place the predator models? I think that it is important to write something about it, either if operator was disturbing or not.

- We agree and have added a few sentences to reflect this in Lines 142-152, 160-163, and 170-182. We conducted the same movements during the pre-stimulus and stimulus periods to avoid affecting the birds’ behaviors as much as possible. For the pre-stimulus period, we walked to the feeding platform (screech owl experiment) or model mount (Cooper’s hawk experiment) and touched it, and for the stimulus period, we walked to the feeding platform/model mount and placed the model on top. Also, the birds in our observation areas are used to observers so although they usually avoid the feeding platform when observers get too close (during both the pre-stimulus and stimulus periods) they rarely leave the area. The birds are typically back to foraging at normal rates from the feeders within seconds of our leaving the immediate area of the feeders.

2. Line 157: how far was the observation place in the screech owl experiment? Was the observer hidden or in plain sight? Please, add these details.

- Thank you for catching this, we have addressed the observation distance and visibility in Lines 142-146 and 161-163.

Results

3. Lines 205-208: so, to test both the “probability of calling” and the “recruitment” hypotheses, you correlated the same things, i.e. number of conspecifics in the flock vs the latency for the first individual to call. What differed was the prediction, right? Aren’t these hypotheses in contrast? When a bird is rare in a mixed flock either could be silent to doesn’t draw attention (probability of calling hypothesis) or be loud to recruit conspecifics (recruitment hypothesis). I suggest to say something about this, also in discussion, if there is some threshold that lead to one or the other hypothesis, and why a rare bird choose to be quite or loud.

- Thank you for this suggestion! Yes, these hypotheses have contrasting predictions. The recruitment hypothesis is more about call rate than volume, but we have added text to the Introduction (Lines 101-113), Methods (Lines 208-212) and Discussion (Lines 362-378) sections to address these comments.

4. Lines 223-225 and 235-242: as I understand, you compared latencies of 2 vs 34 chickadees flocks in the screech owl experiment, latencies of 2 vs 28 chickadees flocks in the hawk experiment, and latencies of 2 vs 28 titmice flocks in the hawk experiment. Therefore, even if you used non-parametric tests, I suggest to be very cautious to draw some conclusions with these results for the oddity effect hypothesis.

- We have adjusted the language to reflect consideration of our small sample size (Lines 275-276, 286, 349-352, and 379-381).

Discussion and Conclusions

5. As I said in the previous comment, because of the small sample size you should be very careful to draw conclusions about the oddity effect hypothesis, as you said as well in conclusions. However, you seemed very convinced of these results (see lines 263-265, 277-279, 282, 299), but I suggest to rephrase using a more cautious tone.

- We have adjusted the language to reflect consideration of our small sample size (Lines 275-276, 286, 349-352, and 379-381), as mentioned earlier.

***Reviewer #2:***

Is the manuscript technically sound, and do the data support the conclusions?

Reviewer #2: Partly

- We thank the reviewer for the comment and hope we have expanded our manuscript to fully meet the requirements for this specification.

Has the statistical analysis been performed appropriately and rigorously?

Reviewer #2: No

- Thank you for this comment, it has been addressed in the updated manuscript in Lines 194-212.

Have the authors made all data underlying the findings in their manuscript fully available?

Reviewer #2: Yes

- We thank the reviewer for the supportive comments!

Is the manuscript presented in an intelligible fashion and written in standard English?

Reviewer #2: No

- We are not sure what the reviewer means by this comment, but would be happy to address any unclear or confusing text in our manuscript.

Review Comments to the Author

General comments: I think this is a nice study, and a good addition to the growing body of literature that is trying to dig into individual behavioral changes in flocking contexts. I do, however, think that the abstract, introduction, and methods could all use some restructuring to make both the question, statistics and field methods clearer. In particular, it is hard to understand exactly what data were taken, whether the data between the two experiments is different, and what the purpose of the two experiments was. I also would strongly consider the authors changing their statistical analysis to take advantage of GLMs and the fact that they can incorporate multiple factors and interactions. I think this reframing in a way that unifies the data and the variability present in the mixed-species flocks may significantly impact the discussion as well.

- We thank the reviewer for the supportive comments on our study and hope that we have adjusted our manuscript to address these concerns!

Abstract:

Line 15 - This first sentence is confusing and, as it stands, could be interpreted multiple ways. I would advise reframing this and maybe swapping the major points of the first two sentences in the abstract. First, describe rarity, then describe its role in predation.

- We have changed the wording to reflect this in Lines 16-19.

Line 25 - The description of the experiment here is also a bit confusing. Given this is the abstract, there isn't much space to explain further, but the details added here don't clarify your findings and instead pose several questions. Why two experiments? How did they differ? Were they analyzed together or separately? Given that (after going through the rest of the paper) the data for the two experiments are different yet are addressing the same question in very similar ways I would just simplify this and state that you had evidence that at least two species (chickadees and titmice) were less likely to call (alarm call?) in response to predators when alone as compared to with conspecifics. I would then maybe use that tiny bit of extra space to explain further the comment below.

- Thank you for the suggestions, we have added clarification in Line 22-23.

Line 27-28 - I also had trouble following this explanation of ruling out alternative hypotheses. Are these your alternatives, or is this what you found? Maybe stating that you tested both the oddity effect and these two alternative hypotheses in one sentence earlier in the abstract would help clarify.

- Thank you for the suggestions, we have added clarification in Lines 26-27 and 30-33.

Introduction:

Line 65 - Somewhere here I think it is important to state that mixed-species groups are often signaling to each other - transmitting information about food or predators etc. This will set up the reasoning below that it isn't always good for specific (rare) individuals to call.

- Thank you for this – we have added some clarification to Lines 50-52.

Line 67 - might be better here to emphasize that signaling can increase predation by making individuals more conspicuous, prior to describing how a reduction in signaling can mitigate this.

- Yes, we have added some clarification to Lines 52-56.

Line 75 - 76 - sentence here about attracting mates and deterring competitors feels out of place. somewhere here, it might be better to lean into the mixed-species group stuff and talk about how there may be benefits to signaling in flocks, but that rarity may interact with appearance/vocal distinctiveness to make it not always beneficial and thus a contributing factor to flock formation, function, and individual behavior

- We have included this suggestion to Lines 66-67.

Line 86 - mixed species should have a dash in it

- Yes, thank you for catching this! This has been added to Line 71.

Line 105 - single conspecifics within heterospecific flocks? And multiple conspecifics within heterospecific flocks? Assuming yes, but this should be clear.

- We have added some clarification to Lines 90-93.

Line 117 - 121 - describing the hypothetical probabilities here seems unnecessary. I would in stead use this space to describe your second alternative hypothesis, perhaps describing the recruitment of birds to mobbing flocks more explicitly. May also expect that birds would call to recruit in order to avoid the oddity effect.

- We have included these suggestions in Lines 105-113.

Methods:

General comment - I'd recommend putting a sentence in near the front of the methods here clearly stating you conducted two experiments using two different predator stimuli. The first you mention that there were two is "Our two experiments involved..."

- We have added some clarification to Line 118.

Also would be helpful to clarify why you used these two different experiments. Is it because these two predators represent different threat levels?

- Yes, this has been added to Lines 130-131.

Line 146 - It is implied, but not clear, that you also used this same plastic screech owl that was used in the other study. Also might be good to state that in the previous work birds responded to it like a predator, as many birds get habituated to plastic stimuli.

- We mention the reactivity of our study species to the screech owl model in Lines 140-141.

Line 152 - are you assuming all the individuals remained in the area following the placement of the stimuli? Did you account for any individual departures driven by the presence of the observer with the stimulus?

- As addressed from Reviewer 1’s suggestions, clarification has been added to Lines 142-152, 160-163, and 170-182, and 180-183 to address pre-stimulus and stimulus bird behavior.

Line 157 - Was the call type recorded? Also, assuming you recorded what species was producing each call.

- Thank you for catching this! The species that called during the 10-minute stimulus period was voice-recorded and this has been clarified in Lines 153-155 and 160-163.

Line 184 - What do you mean by "coded"? Identified? Did you record the time to the first call as well as the number of each call type?

- Yes, we added clarification to Line 184-188. As mentioned in Line 194-195, we recorded the time (latency) to call first, not including the number of each call type.

General method question: Were each of the experiments only conducted once at each of the 36 feeder sites?

- Yes, each experiment was only conducted once at each of the sites. This has also been added to Line 120.

Line 195 - explain how the data were non-normal

- We added some text to address this – Lines 195-199. In s

---

## [Decision Letter · Decision Letter 1]

23 Dec 2025

Thank you for your revisions. Both previous reviewers kindly agreed to read the new version, and both agree that the manuscript is substantially improved. One reviewer noted the somewhat inconsistent way in which the oddity effect hypothesis is formulated, and I agree this aspect should be addressed according to the reviewers' recommendations. Both reviewers also noted a couple of minor formal point to straighten up. I am sure these aspects will take very little time to address, and I look forward to your revised submission.

We look forward to receiving your revised manuscript.

Kind regards,

Valeria Mazza

Academic Editor

PLOS One

Journal Requirements:

Reviewers' comments:

Reviewer's Responses to Questions

**Comments to the Author**

Reviewer #1: All comments have been addressed

Reviewer #2: (No Response)

2. Is the manuscript technically sound, and do the data support the conclusions?

Reviewer #1: Yes

Reviewer #2: Yes

3. Has the statistical analysis been performed appropriately and rigorously?

Reviewer #1: Yes

Reviewer #2: Yes

4. Have the authors made all data underlying the findings in their manuscript fully available?

Reviewer #1: Yes

Reviewer #2: Yes

5. Is the manuscript presented in an intelligible fashion and written in standard English?

Reviewer #1: Yes

Reviewer #2: Yes

Reviewer #1: This version of the manuscript is greatly improved by following reviewers' suggestions. I found only some minor issues that need to be addressed before acceptance:

Line 17: “one” what? Maybe “one individual” or “one bird” or similar is better.

Line 142: “Similar movements” of what/who? Please, replace with something like “Observers moved in the same way” or “We moved in the same way” to be more clear

Line 362: remove the p-value from the conclusion

Nice work!

Reviewer #2: Thank you for addressing my previous recommendations, I think the manuscript is now much stronger and clearer. I do want to apologize for the comment that said the manuscript wasn't presented in an intelligible fashion, that was me clicking incorrectly in the dropdown menu and not realizing I had hit no when I had meant yes.

I only have three minor wording suggestions that you can take or leave and one bigger comment that I strongly recomend you consider (the first comment below). Otherwise I think the manuscript is sound!

Line 53—be careful about defining the oddity effect, it is typically defined by predators being attracted to odd, distinctive prey items within a group, but here you are making it sound like it is about prey behavior. While I don’t fully disagree with the definition, I think it might be better to define the oddity hypothesis in the context of the predator at the start of this paragraph then reframe how you are referring to the prey behavior throughout as a response to oddity or a reduction of risk. For instance, on line 284 you give the example from pigeon predation choice by goshawks and also call it the oddity effect, but this is a very different definition that you present at the start and is more in line with the general literature. I would recommend laying out your terms more clearly and to more appropriately reflect how this phrase is typically used, and make that change in framing throughout the manuscript.

Line 43—Large here is a bit subjective, I’d recommend saying “Groups can often…”

Line 68—Common might be better as “well-known”

Line 257—Thank you for clarifying my confusion about your p-value in the last revision, I should have noticed the effect size. I do however think it might be important to point out that while you didn’t find a significant positive relationship, you instead found a significant negative relationship. It will help to emphasize the finding and also jog the reader to recognize what that finding means, like I needed!

**Do you want your identity to be public for this peer review?** For information about this choice, including consent withdrawal, please see our Privacy Policy

Reviewer #1: No

Reviewer #2: No

---

## [Author Response · Author response to Decision Letter 2]

9 Jan 2026

Response to Reviewers – Plos ONE (PONE-D-25-21798R1)

Tests of the Oddity Effect Hypothesis in mixed-species parid flocks

Dear Dr. Mazza:

Thank you for giving us the opportunity to revise and resubmit our manuscript again. We have copied and pasted the entire set of comments of the reviewers, which we include below. For each

comment, we briefly describe in red text how it was addressed and, if relevant, where

the changes can be found (by line number) in the clean, non-track-changes manuscript.

+++++++++++++++++++++++++

Thank you for submitting your manuscript to PLOS ONE. After careful consideration, we feel that it has merit but does not fully meet PLOS ONE’s publication criteria as it currently stands. Therefore, we invite you to submit a revised version of the manuscript that addresses the points raised during the review process

We thank you and the reviewers for the opportunity to resubmit our manuscript, regardless of the end decision!

Thank you for your revisions. Both previous reviewers kindly agreed to read the new version, and both agree that the manuscript is substantially improved. One reviewer noted the somewhat inconsistent way in which the oddity effect hypothesis is formulated, and I agree this aspect should be addressed according to the reviewers' recommendations. Both reviewers also noted a couple of minor formal point to straighten up. I am sure these aspects will take very little time to address, and I look forward to your revised submission.

Thank you for these comments, they have been addressed in the updated manuscript! Our references have also remained the same.

Comments to the Author:

If the authors have adequately addressed your comments raised in a previous round of review and you feel that this manuscript is now acceptable for publication, you may indicate that here to bypass the “Comments to the Author” section, enter your conflict of interest statement in the “Confidential to Editor” section, and submit your "Accept" recommendation.

Reviewer #1: All comments have been addressed

Reviewer #2: (No Response)

We thank the reviewers for their comments and hope we have expanded our manuscript to fully meet their suggestions.

Is the manuscript technically sound, and do the data support the conclusions?

Reviewer #1: Yes

Reviewer #2: Yes

We thank the reviewers for their comments.

Has the statistical analysis been performed appropriately and rigorously?

Reviewer #1: Yes

Reviewer #2: Yes

We thank the reviewers for their comments.

Have the authors made all data underlying the findings in their manuscript fully available?

Reviewer #1: Yes

Reviewer #2: Yes

We thank the reviewers for their supportive comments!

Is the manuscript presented in an intelligible fashion and written in standard English?

Reviewer #1: Yes

Reviewer #2: Yes

We thank the reviewers for their comments.

Review Comments to the Author

Reviewer #1:

This version of the manuscript is greatly improved by following reviewers' suggestions. I found only some minor issues that need to be addressed before acceptance:

Line 17: “one” what? Maybe “one individual” or “one bird” or similar is better.

Line 142: “Similar movements” of what/who? Please, replace with something like “Observers moved in the same way” or “We moved in the same way” to be more clear

Line 362: remove the p-value from the conclusion

Nice work!

We thank the reviewer for the supportive comments and have addressed these comments in Lines 17, 145-146, and 372.

Reviewer #2:

Thank you for addressing my previous recommendations, I think the manuscript is now much stronger and clearer. I do want to apologize for the comment that said the manuscript wasn't presented in an intelligible fashion, that was me clicking incorrectly in the dropdown menu and not realizing I had hit no when I had meant yes.

I only have three minor wording suggestions that you can take or leave and one bigger comment that I strongly recomend you consider (the first comment below). Otherwise I think the manuscript is sound!

We thank the reviewer for the supportive comments and completely understand the mishap, thank you for your clarification!

Line 53—be careful about defining the oddity effect, it is typically defined by predators being attracted to odd, distinctive prey items within a group, but here you are making it sound like it is about prey behavior. While I don’t fully disagree with the definition, I think it might be better to define the oddity hypothesis in the context of the predator at the start of this paragraph then reframe how you are referring to the prey behavior throughout as a response to oddity or a reduction of risk. For instance, on line 284 you give the example from pigeon predation choice by goshawks and also call it the oddity effect, but this is a very different definition that you present at the start and is more in line with the general literature. I would recommend laying out your terms more clearly and to more appropriately reflect how this phrase is typically used, and make that change in framing throughout the manuscript.

We thank the reviewer for this suggestion and have addressed it in Lines 21-23, 53-57, 99-100, 115-116, 120, 202, 297-298, 314-315.

Line 43—Large here is a bit subjective, I’d recommend saying “Groups can often…”

We agree and have changed the wording in Line 43.

Line 68—Common might be better as “well-known”

We have changed the wording in Line 69.

Line 257—Thank you for clarifying my confusion about your p-value in the last revision, I should have noticed the effect size. I do however think it might be important to point out that while you didn’t find a significant positive relationship, you instead found a significant negative relationship. It will help to emphasize the finding and also jog the reader to recognize what that finding means, like I needed!

We agree and have added this suggestion to Lines 260-263.

---

## [Editor Report · Decision Letter 2]

12 Jan 2026

Tests of the Oddity Effect Hypothesis in mixed-species parid flocks

PONE-D-25-21798R2

Dear Dr. Selman,

thank you for making those changes to your manuscript, which I am now pleased to recommend for publication. I will look forward to seeing your fine contribution in print in due course.

Kind regards,

Valeria Mazza

Academic Editor

PLOS One

---

## [Editor Report · Acceptance letter]

PONE-D-25-21798R2

PLOS One

Dear Dr. Selman,

I'm pleased to inform you that your manuscript has been deemed suitable for publication in PLOS One. Congratulations! Your manuscript is now being handed over to our production team.

Kind regards,

on behalf of

Dr. Valeria Mazza

Academic Editor

PLOS One